bioinformatics/cellular biology

fibrosis, drug repositioning, glaucoma surgery

**Authors for correspondence:**
Stephan Struckmann
e-mail: stephan.struckmann@uni-rostock.de
Georg Fuellen
e-mail: fuellen@uni-rostock.de

†Present address: Omics IT and Data Management Core Facility, German Cancer Research Center (DKFZ), Heidelberg, Germany.

# Suppression of the TGF-β pathway by a macrolide antibiotic decreases fibrotic responses by ocular fibroblasts *in vitro*

Thomas Stahnke[1], Beata Gajda-Deryło[2], Anselm G. Jünemann[1], Oliver Stachs[1], Katharina A. Sterenczak[1], Robert Rejdak[3], Julia Beck[4], Ekkehard Schütz[4], Steffen Möller[2], Israel Barrantes[2], Gregor Warsow[2,†], Stephan Struckmann[2,5] and Georg Fuellen[2]

[1]Department of Ophthalmology, and [2]Institute for Biostatistics and Informatics in Medicine and Ageing Research, Rostock University Medical Center, Rostock, Germany
[3]Department of General Ophthalmology, Medical University in Lublin, Poland
[4]Chronix Biomedical GmbH, Göttingen, Germany
[5]SHIP-KEF, Institute for Community Medicine, Greifswald University Medical Center, Greifswald, Germany

GF, 0000-0002-4994-9829

To elucidate and to inhibit post-surgical fibrotic processes after trabeculectomy in glaucoma therapy, we measured gene expression in a fibrotic cell culture model, based on transforming growth factor TGF-β induction in primary human tenon fibroblasts (hTFs), and used Connectivity Map (CMap) data for drug repositioning. We found that specific molecular mechanisms behind fibrosis are the upregulation of actins, the downregulation of CD34, and the upregulation of inflammatory cytokines such as IL6, IL11 and BMP6. The macrolide antibiotic Josamycin (JM) reverses these molecular mechanisms according to data from the CMap, and we thus tested JM as an inhibitor of fibrosis. JM was first tested for its toxic effects on hTFs, where it showed no influence on cell viability, but inhibited hTF proliferation in a concentration-dependent manner. We then demonstrated that JM suppresses the synthesis of extracellular matrix (ECM) components. In hTFs stimulated with TGF-β1, JM specifically inhibited α-smooth muslce actin expression, suggesting that it

inhibits the transformation of fibroblasts into fibrotic myofibroblasts. In addition, a decrease of components of the ECM such as fibronectin, which is involved in *in vivo* scarring, was observed. We conclude that JM may be a promising candidate for the treatment of fibrosis after glaucoma filtration surgery or drainage device implantation *in vivo*.

# 1. Introduction

Glaucoma is an optic neuropathy accompanied by typical structural and functional defects, such as optic disc damage and loss of vision. These features are shared by both open-angle glaucoma and angle-closure glaucoma [1]. The main risk factor for glaucoma is an increased intraocular pressure which leads to degeneration of retinal ganglion cells [2].

Most glaucoma patients are first treated with anti-hypertensive eye drops [1]. If this treatment is insufficient, surgical procedures are a common second-line option. To enable the drainage of aqueous humour (AH) from the anterior chamber, trabeculectomy and deep sclerectomy are performed [3]. Cyclophotocoagulation is another procedure that reduces AH production [4]. Glaucoma drainage devices, implants that transmit AH from the anterior chamber to the Tenon's space or the subconjunctival space, are gaining acceptance [5]. Many studies were conducted to reduce complications and improve post-operative outcomes of trabeculectomy [6,7]. We recently investigated proteomic data of AH and found an upregulation of pro-fibrotic and inflammatory molecular mediators and of the corresponding putative mechanistic interactions among these [8].

Surgery-induced tissue traumata are physiologically ensuing fibrotic processes of wound healing. Fibroblasts, specifically induced by the transforming growth factor TGF-β, trans-differentiate into myofibroblasts [9], which are characterized by their α-smooth muscle actin (α-SMA) expression and an increased synthesis of extracellular matrix (ECM) compounds, including fibronectin and various types of collagen [10,11]. The overshooting of matrix deposits by myofibroblasts, cellular infiltrates by, e.g. mast cells, and a reduced remodelling of the tissue are called fibrosis or scarring. Many attempts have been made to prevent fibrosis after trabeculectomy to maintain the long-term drainage capacity after surgical intervention. Most commonly, antimetabolites are applied during and after surgery. For example, the cytostatics mitomycin C (MMC) and 5-Fluorouracil (5-FU) are used for this purpose, but they often lead to cell damage and cell death which results in serious side effects [12,13].

The aim of this study was to examine if and how small molecules can inhibit fibrosis after trabeculectomy. The use of small molecules in post-surgical wound healing modulation in glaucoma was recently reviewed [14], but satisfactory solutions are still lacking. Based on an *in silico* drug repositioning effort, we identified the macrolide antibiotic Josamycin (JM) as a candidate inhibitor of fibrosis. This repositioning of a classic antibiotic as an antifibrotic molecule was based on gene expression data. We generated next-generation-sequencing data of human Tenon fibroblasts (hTFs), in which fibrosis was induced by application of TGF-β1. Guided by the transcriptional drug-effect database Connectivity Map (CMap) [15], on a key set of gene/protein interactions, we found JM to feature changes in the opposite direction of the changes of hTF owing to fibrosis. We then confirmed *in vitro* that JM indeed suppresses fibroblast proliferation and fibrotic marker expression in hTFs. We already described this suppression superficially, in a report in German language [16].

# 2. Material and methods

## 2.1. Cell culture

This study was approved by the ethics committee of the University of Rostock (approval ID: A 2011 11) and followed the guidelines of the Declaration of Helsinki. Primary cultures of human Tenon fibroblasts were prepared after child strabismus surgery (Department of Ophthalmology, University of Rostock, Germany) after obtaining of their parent's or legal guardian's informed consent in writing.

Primary hTFs from donors were processed as described in Stahnke *et al.* [10]. Briefly, cells were isolated, allowed to proliferate to a confluent monolayer, trypsinized and subcultured in 25 cm$^2$ cell culture flasks [4]. After reaching a confluent layer in these flasks, the cells were trypsinized again and seeded in 75 cm$^2$ cell culture flasks. For immunofluorescence analysis, cells were seeded on 12 mm glass coverslips (PAA, Cölbe, Germany) and cultured until 60%–70% confluence was reached. For all

analyses, fibroblasts of passages three to five were used. The fibroblastic phenotype was confirmed by immunohistochemistry using a mouse monoclonal anti-vimentin antibody (V2258, Sigma-Aldrich, Munich, Germany) to verify mesenchymal origin and a mouse monoclonal fibroblast-specific surface protein antibody (F4771, Sigma-Aldrich), as previously described [4].

## 2.2. Immunofluorescence

hTFs grown on glass coverslips in multiwell plates were maintained for 24 h under serum-free conditions, followed by the application of increasing concentrations of stimulating TGF-β1 (5 ng ml$^{-1}$, 10 ng ml$^{-1}$ and 20 ng ml$^{-1}$), for 48 h.

The effective concentration of TGF-β1 was determined by observing the induction of α-SMA expression by immunofluorescence. Additionally, the amount of fibronectin, a member of the ECM, was analysed by immunofluorescence.

To study the inhibitory role of JM, stock solution was prepared by dissolving 1 mg of JM in 50 µl of ethanol. For cell culture experiments, JM stock solution was diluted with culture medium (0% foetal calf serum, FCS) to a concentration of 1 mg ml$^{-1}$. This JM solution was applied to the cells in increasing concentrations up to 150 µg ml$^{-1}$. In control experiments, comparable amounts of ethanol without JM were applied to the cells showing no effect on cell viability. After starving in 0% FCS culture medium, cells were incubated with TGF-β1 (10 ng ml$^{-1}$) for 48 h. Thereafter, JM was added and cells were incubated for another 24 h.

After incubation, cells were washed in phosphate buffered saline (PBS) and fixated with 3% paraformaldehyde for 10 min, followed by another three washing steps in PBS and the incubation with the primary antibodies directed against α-SMA (mouse monoclonal anti-α-SMA, Abcam, ab7817) and fibronectin (rabbit polyclonal anti-fibronectin, Abcam, ab2413) in a dilution of 1 : 100, respectively. Thereafter, cells were washed three times with PBS, followed by the incubation with secondary antibodies for 45 min. Secondary antibodies were applied as follows: donkey anti-mouse IgG (H + L)-Alexa Fluor 488 (dilution 1 : 50), or donkey anti-rabbit IgG (H + L)-Cy3 (dilution 1 : 100). After incubation with secondary antibodies, cells were washed again three times with PBS and mounted. Nuclei were stained with 4, 6-diamidino-2-phenylindole (DAPI) (1 µg ml$^{-1}$) included in the mounting medium (Vectashield, Vector Laboratories Ltd., Peterborough, UK). Fluorescent labelling was analysed using a Nikon confocal fluorescence microscope equipped with a digital camera (Nikon Eclipse E400 with D-Eclipse C1, Düsseldorf, Germany). All images depicted in this study were from a single plane through fibroblast cell monolayers equipped with a 40× objective using the same settings at all times.

## 2.3. Cell viability assay

JM was tested for its potential to influence fibroblast viability. Two thousand cells were seeded into each well of a 96-well microtitre plate and incubated in their respective growth medium for 1 day at 37°C, 5% $CO_2$ and 95% relative humidity. Then growth medium was changed to growth medium including JM in a concentration range from 14.89 µg ml$^{-1}$ to 150 µg ml$^{-1}$. After 2 days, cell viability was determined with the CellQuanti-Blue-Assay. Briefly, culture medium was replaced by 200 ml of freshly prepared 10% CellQuanti-Blue reagent (BioAssaySystems, Hayward, CA, USA) in cell culture medium and incubated for another 2 h. Cellular reductase activity was quantified by fluorescence measurements (Fluostar Optima, BMG, Offenburg, Germany). Excitation wavelength was 544 nm, emission wavelength was 590 nm. All measurements were run in quadruplicate.

## 2.4. Bromodeoxyuridine cell proliferation assay

JM was also tested for its ability to arrest fibroblast proliferation. Two thousand cells were seeded into each well of a 96-well microtitre plate in growth medium and incubated under standard conditions for 1 day. Then growth medium was exchanged with growth medium including JM in a concentration range from 14.9 µg ml$^{-1}$ to 150 µg ml$^{-1}$, respectively. After a 32 h incubation period, bromodeoxyuridine (BrdU) (chemiluminescence) (Roche Diagnostics, 173 Mannheim, Germany) (100 mM) was added to each well and incubation continued for additional 16 h. BrdU incorporation into DNA was measured according to the supplier's instructions of the Cell Proliferation ELISA, BrdU. Cells not subjected to JM served as negative control. All measurements were run in quadruplicate.

## 2.5. Western blot analysis

For Western blot analysis, primary hTFs treated in the same manner as described above were washed with PBS and scraped off into Laemmli buffer containing 1% sodium dodecyl sulfate (SDS). All samples were boiled for 10 min. For separation by SDS-PAGE, total cellular extracts (10–30 µg of protein per lane) were loaded in 7.5% or 10% polyacrylamide gels for larger and smaller proteins, respectively. By immunoblotting, proteins were transferred to polyvinylidene fluoride membranes (0.2 µm, BIO-RAD, Munich, Germany). After blocking the membranes with 5% non-fat dry milk powder in TRIS-buffered saline (TBS) for 30 min, blots were incubated with individual primary antibodies at 4°C overnight. The following primary antibodies were used (dilutions are given in brackets): mouse monoclonal anti-collagen I, Abcam, ab6308 (1 : 500), mouse monoclonal anti-collagen VI, Abcam, ab 78504, mouse monoclonal anti-fibronectin, Sigma Aldrich, F7387 (1 : 500), mouse monoclonal anti-β-tubulin, Sigma Aldrich, T5293 (1 : 250), mouse monoclonal anti-β-actin, Sigma Aldrich, A2228 (1 : 500), mouse monoclonal anti-vimentin, Sigma Aldrich, V2258 (1 : 500) and mouse monoclonal anti-α-SMA, Abcam, ab7817 (1 : 500). After three washing steps with TBS including Tween 20, membranes were incubated with secondary horseradish peroxidase-conjugated anti-mouse, GE Healthcare (Amersham; Buckinghamshire, UK) NXA931 (1 : 2,500) or anti-rabbit, BIO-RAD (Munich, Germany) 170–6515 (1 : 2,500) IgG. Visualization of bound antibodies was performed by the enhanced chemiluminescence procedure as described by the manufacturer (Thermo scientific, Pierce, Rockford, USA). Optical density of each band was normalized to the corresponding β-tubulin band. The blots were quantified using IMAGEJ software based on guidelines given by Gassmann *et al.* [17].

## 2.6. RNA extraction

RNA was isolated from harvested cells by TRIzol (Life Technologies GmbH, Darmstadt, Germany) reagent. Briefly, after the addition of 10% chloroform (Baker, Deventer, The Netherlands), cells were vortex mixed and purified RNA was isolated in two centrifugation steps. The aqueous supernatant was adjusted to 35% ethanol and loaded onto an RNeasy column (QIAGEN, Hilden, Germany). After washing the column, residual DNA was digested with DNase I (QIAGEN) treatment followed by three washing steps. Total RNA was eluted with 100 µl of sterile, RNase-free water.

## 2.7. Statistical analysis

Statistical analyses of the experimental data (that is, excluding the gene expression data) were performed using GRAPHPAD PRISM 5 data analysis software (GraphPad Software, La Jolla, USA). Significance tests of different culture conditions were conducted using one-way ANOVA. Differences were considered as statistically significant for $p$-values < 0.05. Bar charts were generated using mean and standard deviation (s.d.).

## 2.8. Next-generation-sequencing processing and analyses

The primary hTF fibrotic cell culture model was used to generate gene expression datasets by next-generation-sequencing (NGS; Chronix Biomedical GmbH, Göttingen, Germany) for two types of samples, that is, 'non-fibrotic' (control) hTF cells' and 'fibrotic' hTF cells, the latter obtained by applying TGF-β1.

Poly-A RNA was enriched from 1 µg of total RNA and cDNA sequencing libraries were prepared using the NEBNext Ultra RNA Library Prep Kit for Illumina. Libraries were sequenced on a NextSeq500 platform (Illumina) to obtain approximately 50 million 75-base-single-end reads per library. The raw data are available at https://www.ebi.ac.uk/ena/data/view/PRJEB38998. Low-quality reads were identified by FASTQC-0.11.2 and were filtered using TRIMMOMATIC-0.32. The remaining reads were mapped to the Homo sapiens Ensembl GRCh38 assembly using TOPHAT2 and BOWTIE2 with default parameters. Finally, HTSeq-count-0.6.1 (part of the SAMtools-1.3246) was used to generate the count files based on the mapped reads.

The JM gene expression values were derived from the NGS counts using reads per kilobase of transcript, per million mapped reads (RPKM) [18]. Differential gene expression data were then obtained as log-fold changes contrasting the gene expression of fibrotic cells with control cells.

## 2.9. Gene expression analysis and scoring of relevant gene/protein interactions

The CMap [15] raw data were reanalysed and corrected for batch effects by established protocols. A script performing that process is available from https://bitbucket.org/ibima/cmapreanalysed2016. The script has meanwhile been developed into an R package and evaluated for correlation-based drug *in silico*

screening [19]. The data describe the effect of 1187 drugs in a total of 3395 CMap experiments, where one CMap experiment is defined as the application of a drug in a specific concentration to a specific cell line. Differential gene expression data were then obtained as log-fold changes contrasting the gene expression before and after drug application.

To systematically identify drugs from the CMap that target mechanisms behind fibrosis, the FocusHeuristics [20] was applied. This method was developed to integrate a network with gene expression data, and filtering the gene/protein interactions (links) in the network along which the 'most relevant' gene expression changes took place. More specifically, based on a set of scores that defines 'relevance' as described next, the FocusHeuristics calculates a set of highest-scoring links between the genes/proteins in the network. Here, we used the human STRING (version 10) protein-protein-interaction network consisting of 990 338 edges and 17 315 nodes, which we constructed using the R packages STRINGdb [21] and igraph, using all STRING edges, that is, without applying a STRING score threshold. For filtering, we instead used the FocusHeuristics with a threshold of 0.01% for all of its three built-in scores, reflecting the 'relevance' of the interactions/links between genes/proteins in the network based on the following criteria:

 (i) differential expression of the pairwise interacting genes;
 (ii) joint high expression of the pairwise interacting genes; and
 (iii) single-gene fold changes.

Specifically, we used the 0.01% threshold for:

 (i) the LinkScore, separately for the upper and lower quantiles, selecting the 0.01% most-upregulated and the 0.01% most-downregulated links, where the LinkScore is the sum of the log-fold changes of the two genes connected by an activating interaction (stimulation), and the difference of these same log-fold changes for an inhibition;
 (ii) the InteractionScore, again separately for the upper and lower quantiles, where the InteractionScore is the smaller of the two sums of the log-fold changes of the two conditions to be compared, summing up the expression levels of the two genes forming the interaction, separately considering each condition; and
 (iii) for the log-fold changes of the two genes connected by the interaction.

For each interaction/link, the three scores are considered one after the other for filtering [20].

## 2.10. Comparative analysis of changes to gene expression to identify candidates for repositioning

As described, to obtain the log-fold changes (that is, the differential gene expression data), we first contrasted the gene expression of fibrotic cells with control cells (that is, with and without induction of fibrosis by TGF-β1), and filtered the set of highest-scoring links describing the fibrotic effect using the FocusHeuristics. Secondly, the same FocusHeuristics link filtering was done based on the CMap gene expression values of the single drugs, considering different concentrations of the same drug as separate experiments. That is, for each of the total of 3395 CMap experiments to be considered, we contrasted the drug with its respective control, defining sets of highest-scoring links describing the drug effects. We always used default parameters for the FocusHeuristics, overall yielding one set of links describing fibrosis as well as 3395 sets of links describing drug effects. Because we were interested in reversing the fibrosis-related gene expression changes, we inverted the LinkScores for the fibrosis effects, so that we could directly compare them to the LinkScores for the drug effects.

For each drug, the set of links selected by the FocusHeuristics for the drug was then compared to the set of links selected by the FocusHeuristics for describing fibrosis. Specifically, the Jaccard index (that is, the degree of overlap using set theory [22]) between the links describing the drug effect and the links describing the fibrosis effect was computed. Similarly, the Pearson correlation of the LinkScores describing the drug effect with the LinkScores describing the fibrosis effect was calculated. Here, we considered the union set of all selected links, so that the correlation could be calculated for all links for which at least one LinkScore is above the threshold used for the FocusHeuristics.

## 2.11. Visualization of putative antifibrotic mechanisms

Finally, a differential network analysis was performed comparing fibrotic cells with control cells using *ExprEssence* [23], with default parameters, revealing and visualizing the molecular mechanisms behind

**Figure 1.** (*a*) *ExprEssence*-based link set describing fibrosis, based on differential expression of the fibrotic versus control cells. From the full set of links, the top-47 links, scoring highest by the absolute value of the *ExprEssence* LinkScore, are visualized using the Cytoscape 2.8 plugin *ExprEssence*. The subnetworks formed by the links are ordered by size. (*b*) For the same set of links, the changes triggered by JM are visualized. In both plots, colour mapping was done continuously from the highest (red) to the lowest value (green), centered at white (for an average normalized expression value of 1.68, and for a LinkScore of 0, respectively).

TGF-β1-induced fibrosis (figure 1). *ExprEssence* employs the LinkScore (LS) formula for scoring interactions, and was used to visualize the network of the 47 links most regulated by fibrosis. For the same set of 47 links, the effect of JM was visualized head-to-head based on the same genes, employing again the LS formula. We used a colour code with a continuous interpolation between green (downregulation, LS: -0.5), white (no change, LS: 0) and red (upregulation, LS: 0.5).

# 3. Results

## 3.1. Identification of JM as a candidate antifibrotic compound by repositioning

Based on our established model of fibrosis in primary human tenon fibroblasts, the *in silico* drug screening identified small molecules from the CMap database that may reverse the gene expression changes caused by fibrosis (for details, see *Methods*). After filtering the expression data using the network-based FocusHeuristics tool, we thus correlated the inverted interaction/link scores based on gene expression in fibrosis to the interaction/link scores of drugs in CMap, using Pearson correlation and the Jaccard index. For Pearson correlation, the drugs with the highest correlations are listed in table 1. Comparing to data generated as part of the cTRAP project (https://bioconductor.riken.jp/packages/3.8/bioc/vignettes/ cTRAP/inst/doc/comparing_DGE_with_perturbations.html), we see that Pearson correlations between 0.1 and 0.2 are commonly found in analyses of CMap-related data; for example, in the cTRAP manuscript, ENCODE-based gene expression data describing a genetic perturbation were used to retrieve the same perturbation from the CMap follow-up project [24], with a Pearson correlation of 0.193 for the best match. For the Jaccard index, the drugs with the highest correlations are listed in table 2.

We thus found JM ranked best by Pearson correlation and Jaccard index. For the next-ranking compounds, we found some more overlap between the two rankings based on these two similarity measures, that is, iobenguane and heliotrine, but we did not investigate these next-ranking compounds any further.

## 3.2. Identification of putative antifibrotic mechanisms of JM

To identify the molecular mechanisms underlying the antifibrotic effects of JM, a differential regulatory network visualization contrasting fibrotic and non-fibrotic cells was conducted using *ExprEssence* [23], revealing a set of mechanisms that may be involved in fibrosis, and in its inhibition by JM (figure 1).

**Table 1.** Ranking of potential antifibrotics based on Pearson correlation of the CMap-based differential gene expression data of compound effects represented by LinkScores, compared to differential gene expression data for the fibrotic effect represented by LinkScores. (A CMap entry consists of drug name, concentration applied, and cell line treated.)

| rank | CMap drug | $r^2$ |
|---|---|---|
| 1 | josamycin_4.8e-06_MCF7 | 0.13581 |
| 2 | meticrane_1.46e-05_HL60 | 0.10036 |
| 3 | prazosin_9.6e-06_MCF7 | 0.09314 |
| 4 | prilocaine_1.56e-05_HL60 | 0.09299 |
| 5 | heliotrine_1.28e-05_HL60 | 0.07841 |
| 6 | benzamil_1.12e-05_HL60 | 0.07567 |
| 7 | mifepristone_9.4e-06_PC3 | 0.07013 |
| 8 | fenbufen_1.58e-05_HL60 | 0.05239 |
| 9 | naproxen_1.74e-05_HL60 | 0.05227 |
| 10 | Prestwick-642_1.38e-05_MCF7 | 0.05017 |
| 11 | lovastatin_9.8e-06_MCF7 | 0.04775 |
| 12 | iobenguane_1.08e-05_MCF7 | 0.04671 |
| 13 | etilefrine_1.84e-05_MCF7 | 0.04502 |
| 14 | glafenine_9.8e-06_MCF7 | 0.04457 |
| 15 | oxybenzone_1.76e-05_PC3 | 0.04357 |
| 16 | flucloxacillin_8.4e-06_MCF7 | 0.04316 |
| 17 | terguride_1.18e-05_HL60 | 0.04153 |
| 18 | Prestwick-1080_1.5e-05_MCF7 | 0.04044 |
| 19 | minaprine_1.08e-05_MCF7 | 0.03852 |
| 20 | antazoline_1.32e-05_MCF7 | 0.03806 |

**Table 2.** Ranking for potential antifibrotics based on the Jaccard index of the links describing the fibrotic effect and the links describing the compound effects. (From rank 15 onwards, the intersection is empty and the Jaccard index is zero.)

| rank | drug | Jaccard |
|---|---|---|
| 1 | josamycin_4.8e-06_MCF7 | 0.01117 |
| 2 | piretanide_1.1e-05_MCF7 | 0.01117 |
| 3 | dihydroergocristine_5.6e-06_MCF7 | 0.00555 |
| 4 | iobenguane_1.08e-05_MCF7 | 0.00555 |
| 5 | estradiol_1,00E-07_HL60 | 0.00555 |
| 6 | novobiocin_6.4e-06_MCF7 | 0.00555 |
| 7 | alprenolol_1.4e-05_MCF7 | 0.00555 |
| 8 | fluorocurarine_1.16e-05_MCF7 | 0.00555 |
| 9 | calcium pantothenate_8,00E-06_MCF7 | 0.00555 |
| 10 | Prestwick-665_1.16e-05_HL60 | 0.00555 |
| 11 | heliotrine_1.28e-05_HL60 | 0.00555 |
| 12 | alcuronium chloride_5.4e-06_MCF7 | 0.00555 |
| 13 | Prestwick-1103_1.96e-05_HL60 | 0.00555 |
| 14 | naftidrofuryl_8.4e-06_MCF7 | 0.00555 |
| 15 | paracetamol_2.64e-05_PC3 | 0.00555 |

In figure 1a, the largest subnetwork is centred around ACTC1, the α-actin found most prominently in cardiac muscle, the normalized gene expression of which is upregulated from 0.29 to 2.63. Upregulation of its STRING interaction partners, among them ACTA2 (α-SMA, from 2.63 to 3.40) and ACTG2 (γ2-SMA, from 1.59 to 2.96), suggests the upregulation of their mutual interactions. Overall, an increased expression rate of the actin cytoskeleton, and in particular α-SMA, indicates a transformation of fibroblasts to myofibroblasts, which are also characterized by an increased synthesis of ECM components [10,11]. The second-largest subnetwork centres around CD34, which is downregulated from 2.37 to 0.75. Downregulation of CD34 was previously found to indicate fibrocyte to myofibroblast differentiation [25], and CD34 expression was found to correlate with the degree of fibrosis in continuous ambulatory peritoneal dialysis patients [26]. Recently, it was found to decrease, together with its interaction partners CDH5 and KIT, in fibrotic versus non-fibrotic conjunctival fibroblast cell lines from patients with and without previous glaucoma surgery, respectively [27] (their fig. 4 and electronic supplementary material, table S9). The third-largest subnetwork features the upregulation of IL11, IL6 and BMP6, known to trigger fibrosis (myofibroblast cell fate or scar tissue) in hTFs [28,29]. IL6 is one of the most often implicated mediators of fibrosis; its upregulation was also found in the fibrotic versus non-fibrotic patient cell lines [27]. Furthermore, IL11 was recently implicated in cardiovascular and lung fibrosis [30,31]. The JM data (figure 1b) suggest the reverse regulation of almost all interactions constituting the three largest subnetworks. Notably, in the first subnetwork, the interaction ACTC1–CORO2A is reversed the most, followed by IL6 - BMP6 in the third subnetwork. The key publication for CORO2A [32] reflects our overall observation that fibrosis is triggered by inflammatory processes, with consequences for the ECM; the specific hypothesis here is that Coronin2A mediates these consequences, and that JM acts in part by triggering its downregulation. The two interactions discussed here are exactly the ones that are contributing to the Jaccard index in table 2.

Inspecting the runner-up subnetworks, we find that the fourth-largest one centres around cartilage oligomeric matrix protein (COMP) and other ECM proteins such as MATN3, which are upregulated, reflecting fibrotic ECM deposition. Concordantly, COMP was suggested to foster skin, lung and liver fibrosis [33–35]. Surprisingly, however, COMP was found downregulated in the data from the fibrotic versus non-fibrotic patient cell lines [27]; there is no further explanation given in [27]. The fifth-largest subnetwork centres around PRKG2, for which there is no known role in fibrosis. For the two runner-up subnetworks of unclear significance, the reverse regulation by JM is far less pronounced than for the top three subnetworks.

The remaining subnetworks (of size smaller than five) feature genes involved in a variety of biological processes, and they are expected to include spurious findings with a higher likelihood than the larger subnetworks. The most notable observation is the downregulation of collagens 13A1, 14A1 and 21A1, which is in fact reversed by JM, as is the downregulation of collagen 4A1 in the fourth-largest subnetwork. This reversal by JM holds for the majority of the up- and downregulations in the subnetworks of size smaller than 5.

In sum, based on the consistent rankings of JM as the best-reversing compound of Tenon-fibroblast-specific fibrosis among all CMap compounds, and the confirmation of plausible antifibrotic mechanisms of JM by the *ExprEssence* visualization, we decided to investigate it as a potential antifibrotic.

## 3.3. Immunofluorescence studies of Tenon fibroblasts and JM effects on Tenon fibroblasts

After stimulation of hTFs with TGF-β1, a concentration-dependent increase of α-SMA was found, demonstrating fibroblast transformation into myofibroblasts (figure 2a). Also, the expression of the ECM protein fibronectin increased considerably. Quantification by IMAGEJ software of the fluorescence signal density confirmed these results (figure 2b,c).

Cytotoxicity of JM was tested on the hTFs using the CellQuanti-Blue assay to measure cell viability after treatment. Fibroblasts were cultured *in vitro* and treated with JM over a wide concentration range (figure 3). JM has only a negligible influence on cell viability at the highest concentration tested; lower concentrations do not affect cell viability. By contrast, JM significantly reduced the cell proliferation of hTFs in a concentration-dependent manner, as shown by BrdU assay.

The incubation of hTFs with increasing doses of JM (25 µg ml$^{-1}$, 50 µg ml$^{-1}$) for 24 h resulted in a concentration-dependent decrease of fibronectin expression, demonstrating an inhibitory effect on ECM synthesis (figure 4a).

The combination of TGF-β1 stimulation and application of JM resulted in a decrease in fibronectin and α-SMA expression, compared to control cells stimulated by TGF-β1 alone, however the decrease

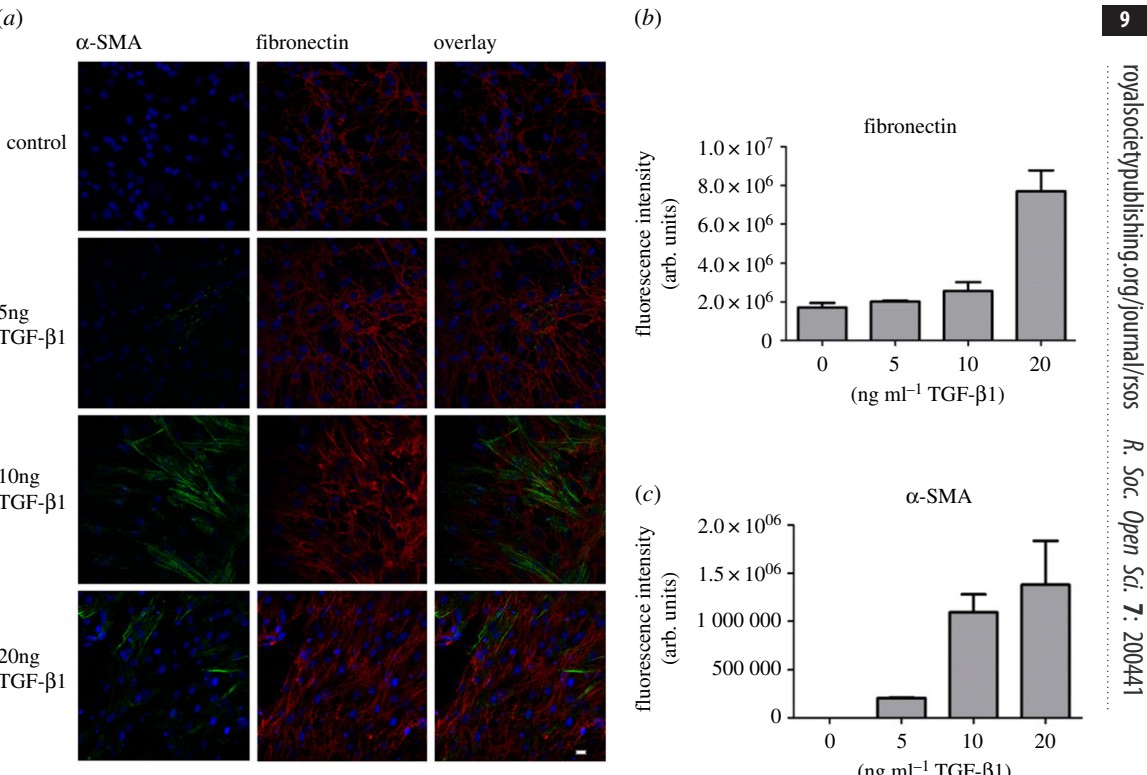

**Figure 2.** (*a*) Immunofluorescence of α-SMA (green) and fibronectin (red) in hTFs treated with increasing concentrations of TGF-β1. Nuclei were stained by DAPI included in the mounting medium. Bar represents 20 µm. (*b*) Quantification of fibronectin expression after TGF-β1 stimulation in hTFs. (*c*) Quantification of TGF-β1 induced α-SMA expression in hTFs.

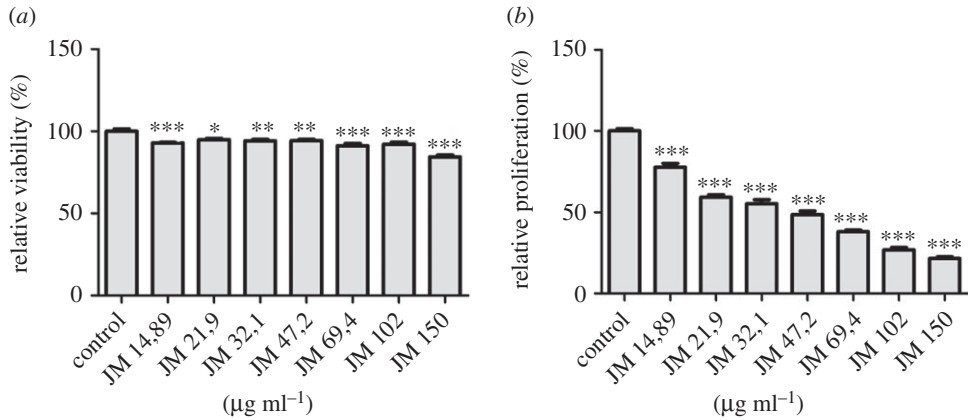

**Figure 3.** Relative viability (*a*) and relative proliferation (*b*) of hTFs in response to JM *in vitro*. hTFs were treated with JM in concentrations as indicated. CellQuanti-Blue-assay was carried out to quantify relative cell viability. Cell proliferation ELISA, BrdU (chemiluminescence)-assay was carried out to quantify relative proliferation. Data are presented as mean ± s.d. The results represent the means of three independent experiments. Levels of significance: $^{*}p \leq 0.05$; $^{**}p \leq 0.01$; $^{***}p \leq 0.001$.

was not statistically significant, only a tendency could be observed. Quantification of these results using IMAGEJ software is shown in figure 4*b*,*c*.

Based on our cell viability data, higher concentrations of JM up to 150 µg ml$^{-1}$ were used to increase the inhibitory effect on fibrotic ECM expression in hTFs. These higher concentrations resulted in a more intense decrease of fibronectin and α-SMA expression in cells without (figure 5) and in cells with TGF-β1 stimulation (figure 6). The combination of TGF-β1 stimulation and application of JM resulted in a statistically significant decrease in both fibronectin and α-SMA expression, compared to cells stimulated by TGF-β1 alone. Cell death or apoptotic nuclei were not observed.

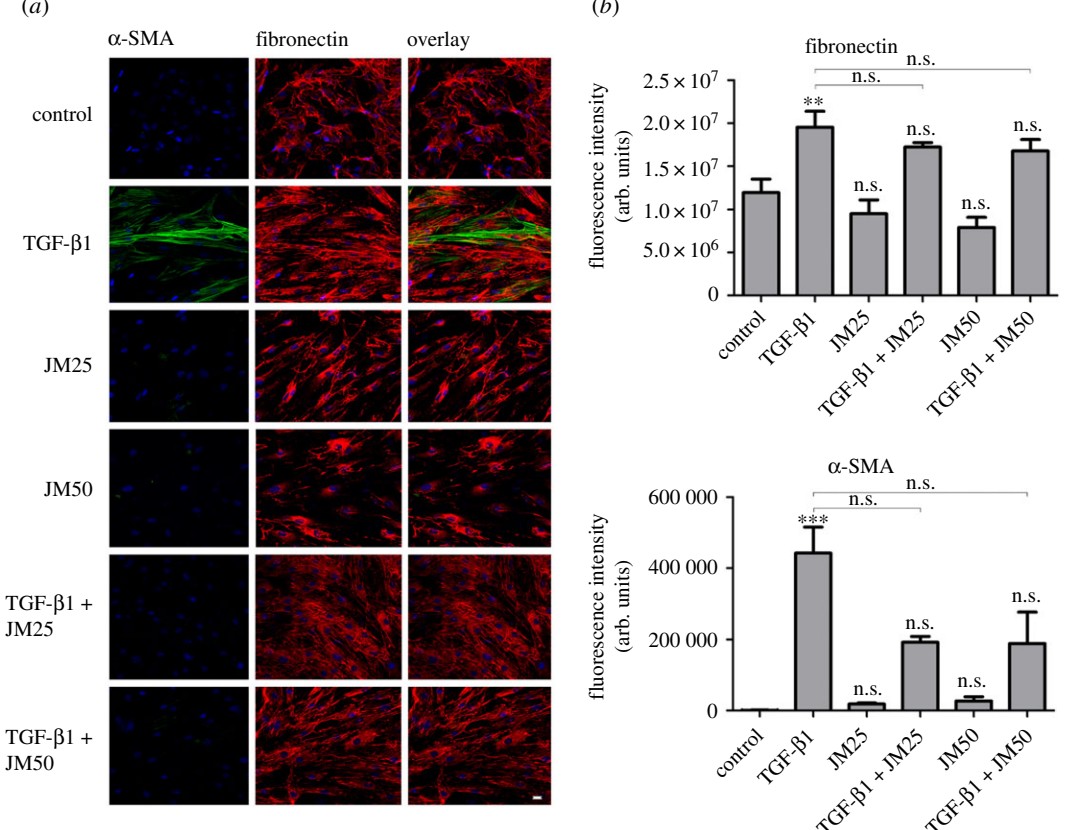

**Figure 4.** (*a*) Immunofluorescence of α-SMA (green) and fibronectin (red) in hTFs treated with TGF-β1 (10 ng ml$^{-1}$), JM25 (25 μg ml$^{-1}$, 30 × 10$^{-9}$ mol ml$^{-1}$), JM50 (50 μg ml$^{-1}$, 60 × 10$^{-9}$ mol ml$^{-1}$), TGF-β1 + JM25, and TGF-β1 + JM50. Bar represents 20 μm. (*b*) Quantification of fibronectin expression after incubating with TGF-β1, JM25, JM50, TGF-β1 + JM25, and TGF-β1 + JM50 in hTFs. (*c*) Quantification of α-SMA expression after incubating with TGF-β1, JM25, JM50, TGF-β1 + JM25 and TGF-β1 + JM50 in hTFs. Each column represents the mean ± s.d. from three independent experiments. The asterisk (*) in (*b*) and (*c*) indicates significance obtained by comparison of TGF-β1, JM25, TGF-β1 + JM25, JM50 and TGF-β1 + JM50 to untreated cultures. Levels of significance for (*b*) and (*c*): *$p \leq 0.05$; **$p \leq 0.01$; ***$p \leq 0.001$.

In order to compare the amount of various ECM components, cell lysates were quantified by immunoblotting. As loading control, β-tubulin was quantified, showing no notable differences in signal intensity underlining equal amounts of loaded proteins. In hTFs cultures stimulated with TGF-β1 when compared to untreated controls, an increase in the amounts of fibronectin was observed, as well as of collagen I and α-SMA. If hTFs were treated with JM alone in increasing concentrations, a decrease in the respective proteins could be observed. The combined application of TGF-β1 and JM in the same, increasing concentrations resulted in a concentration-dependent decrease of fibronectin, α-SMA, collagen I and collagen VI expression (figure 7*a*). We observed statistically significant decrease of α-SMA expression for TGF-β1 + JM at concentrations of 75 μg ml$^{-1}$ and 150 μg ml$^{-1}$ and also of collagen I for TGF-β1 + JM at a concentration of 150 μg ml$^{-1}$. The cytoskeletal proteins vimentin and β-actin were not affected. Mitomycin C was also applied for comparison. Quantification of Western blot analyses is shown in figure 7*b–e*.

## 4. Discussion

The aim of our study was to discover a small molecule that can inhibit the process of fibrosis in ocular fibroblasts, based on the analysis of gene expression data, and to validate our findings *in vitro*. Glaucoma surgery leads to tissue trauma by inducing an inflammatory response. Activation of mediators such as TGF-β1 and VEGF-A promote the migration of polymorphonuclear cells and macrophages, removes debris, prevents infection and promotes angiogenesis. TGF-β in particular induces transdifferentiation of fibroblasts to myofibroblasts, which produce α-SMA and also promote expression of ECM proteins

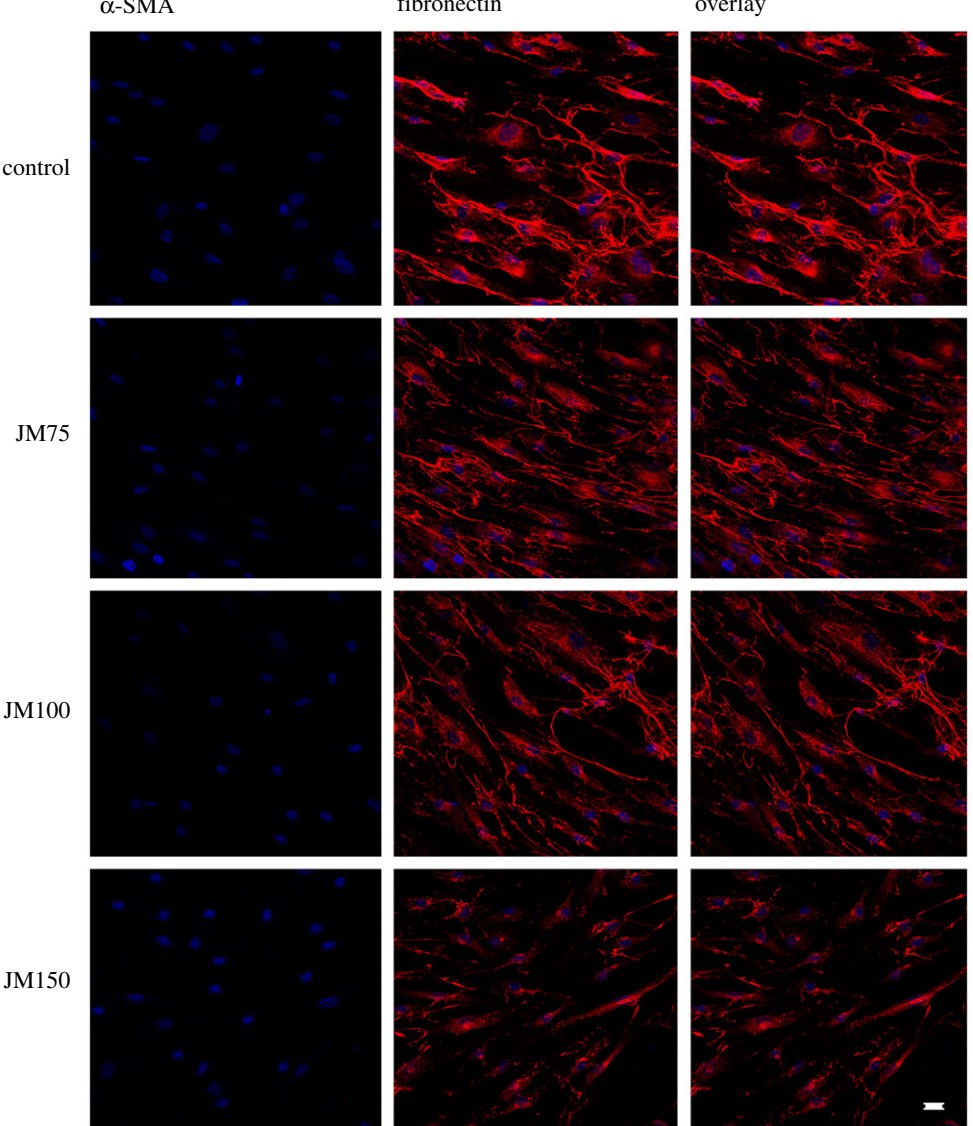

**Figure 5.** Immunofluorescence of α-SMA (green) and fibronectin (red) in hTFs treated with JM75 (75 µg ml$^{-1}$, 90*10$^{-9}$ mol ml$^{-1}$), JM100 (100 µg ml$^{-1}$, 120 × 10$^{-9}$ mol ml$^{-1}$), and JM150 (150 µg ml$^{-1}$, 180 × 10$^{-9}$ mol ml$^{-1}$) concentrations. Bar represents 20 µm.

(fibronectin and members of the collagen family). Persisting presence of myofibroblasts leads to excessive scar formation [36]. MMC and 5-FU are commonly used to prevent scarring after trabeculectomy surgery proving to be efficient in many studies [37,38], but often leading to hypotony with maculopathy, bleb infection, thin vascular blebs and endophthalmitis, which are potentially causing vision loss [39,40]. Some studies examine the efficacy of Bevacizumab, an anti-VEGF drug, alone or together with MMC against post-surgical fibrosis. These experiments showed that combination of both substances did not cause more benefit than using MMC alone [41]. Also, Bevacizumab alone was not more efficient than MMC alone in inhibiting fibrosis after trabeculectomy [42].

Another approach to decrease post-surgical fibrosis is the use of small molecules which counteract with signalling pathways relevant for ECM component deposition. One of these small molecules is pirfenidone, which was recently demonstrated to attenuate TGF-β1 induced expression of fibronectin and α-SMA in ocular fibroblast cultures, without impairing cell viability [10]. In our exploration of antifibrotic drugs using *in silico* drug screening from the CMap database, pirfenidone was not listed as one of the most effective agents against fibrosis. Instead, we identified the antibiotic macrolide JM as a potentially most effective antifibrotic.

Hence, we examined JM, a 16-membered ring macrolide, as a potential antifibrotic small molecule. As an antibiotic, it was reported to be effective against *Streptococcus pyogenes* infections [43] and bronchopulmonary infections [44]. Macrolides were examined in various studies that suggested an

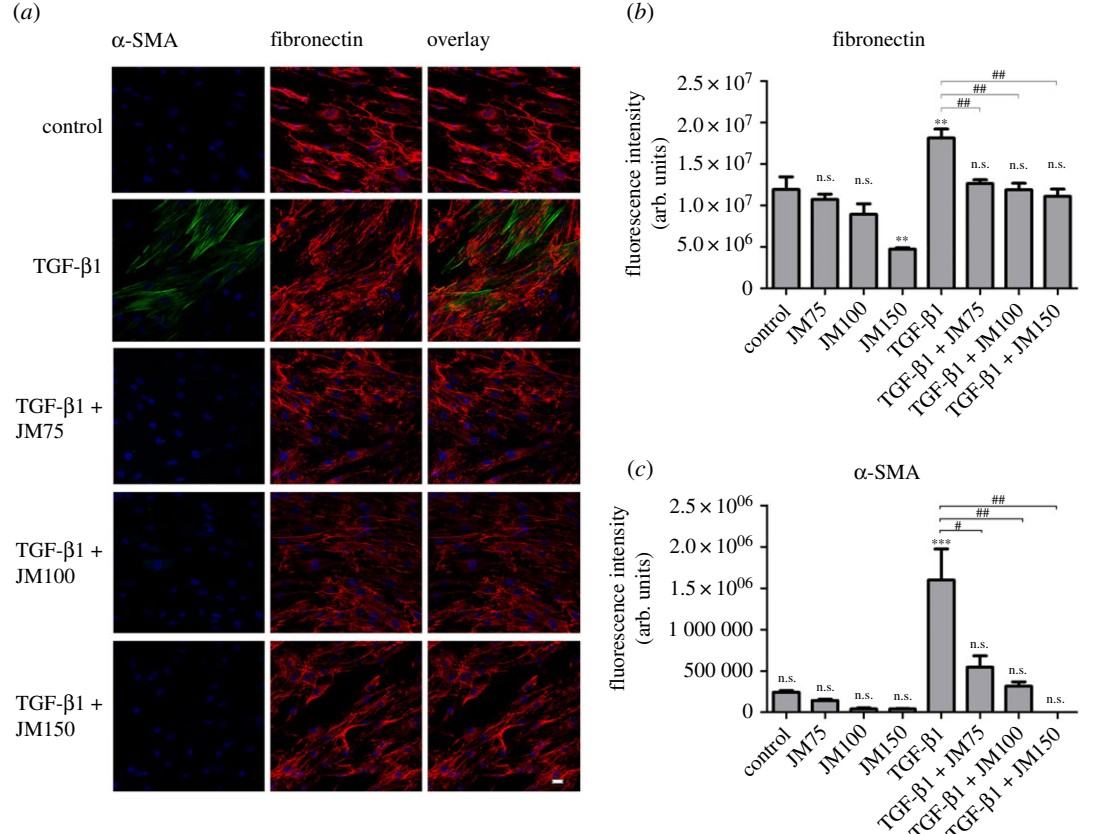

**Figure 6.** (*a*) Immunofluorescence of α-SMA (green) and fibronectin (red) in hTFs treated with TGF-β1 (10 ng ml$^{-1}$), TGF-β1 + JM75, TGF-β1 + JM100, and TGF-β1 + JM150. Bar represents 20 μm. (*b*) Quantification of fibronectin expression after incubating with JM75, JM100, JM150 (figure 5), TGF-β1, TGF-β1 + JM75, TGF-β1 + JM100 and TGF-β1 + JM150 in hTFs. (*c*) Quantification of α-SMA expression after incubating with JM75, JM100, JM150 (figure 5), TGF-β1, TGF-β1 + JM75, TGF-β1 + JM100 and TGF-β1 + JM150 in hTFs. Each column represents the mean ± s.d. from three independent experiments. The asterisk (*) in (*b*) and (*c*) indicates statistical significance as obtained by comparison of TGF-β1, JM75, JM100, JM150, TGF-β1 + JM75, TGF-β1 + JM100 and TGF-β1 + JM150 to untreated cultures. The hash sign (#) in (*b*) and (*c*) indicates significances obtained by comparison of TGF-β1-treated cultures and cultures with combined treatment TGF-β1 + JM75, TGF-β1 + JM100 and TGF-β1 + JM50. Levels of significance: *(#) $p \leq 0.05$; **(##) $p \leq 0.01$; ***(###) $p \leq 0.001$.

impact on fibrotic processes. For example, Kanai *et al.* examined the influence of two macrolides, roxithromycin (RXM) and JM on matrix metalloproteinase (MMP) production from nasal polyp fibroblasts *in vitro*. They observed that addition of RXM, but not JM, suppressed production of MMPs [45]. Effects of RXM in inhibiting fibroblasts growth on nasal polyp fibroblasts were also confirmed in other studies [46,47]. It was also confirmed that some of the 14-membered ring macrolides, including RXM, have an antiangiogenic and anti-tumor effect [48]. In other studies, the macrolide JM was identified to be an effective inhibitor of T-cell proliferation by inhibiting their synthesis of IL-2 [49]. IL-2 synthesis by myofibroblasts (activated fibroblasts) was already reported to be increased in patients suffering from post-radiation fibrosis [50]. Dysregulation of IL-2 levels in T lymphocytes was also observed in patients with cystic fibrosis [51], suggesting that this cytokine plays a role in fibrotic processes.

Based on our *in silico* repositioning results, we chose to investigate the antifibrotic activity of JM in ocular fibroblasts, motivating it as a candidate for inhibiting post-surgical fibrosis in trabeculectomy. In a recent study, it was confirmed in gene expression experiments based on patient-based cell lines (from patients with or without previous glaucoma surgery) that the increased production of IL-6 and the decreased production of CD34 by fibroblasts is a key to understand fibrotic processes after glaucoma surgery [27]. Patients were re-operated based on failure of the previous glaucoma surgery owing to fibrosis, indicating that our fibrosis model reflects at least in part the *in vivo* process of fibrosis after glaucoma surgery. Moreover, our fibrotic cell culture model, based on hTFs stimulated by TGF-β1, was already used extensively [10]. This model system was the basis of all experiments, the ones yielding the prediction of the antibiotic, as well as the confirmatory ones.

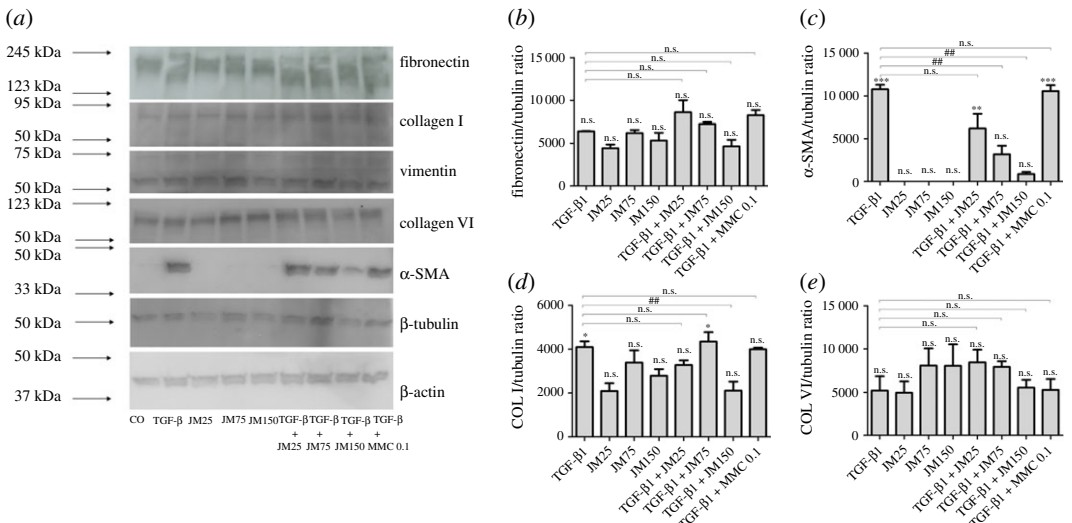

**Figure 7.** (a) Western blot analysis of cell lysates of hTFs under different culture conditions, using antibodies against fibronectin, collagen I, vimentin, collagen VI, α-SMA and β-actin; β-tubulin was used as a loading control. kDa are marked on the left, as indicated. (b–e) Quantification of Western blot data of fibronectin (b), α-SMA (c), collagen I (d) and collagen VI (e) for hTFs. Each column represents the mean ± s.d. from three independent experiments. The asterisk (*) indicates significance obtained by comparison of TGF-β1, JM25, JM75, JM50, TGF-β1 + JM25, TGF-β1 + JM75 and TGF-β1 + JM150 to untreated cultures. The hash sign (#) indicates significance obtained by comparison of TGF-β1-treated cultures and cultures with combined treatment TGF-β1 + JM25, TGF-β1 + JM75, and TGF-β1 + JM150. Levels of significance: *(#) $p \leq 0.05$; **(##) $p \leq 0.01$; ***(###) $p \leq 0.001$.

In the immunofluorescence experiments that we conducted, we focused on fibronectin and α-SMA, which are both synthesized by myofibroblasts [52]. α-SMA is an indicator for myofibroblasts, whereas synthesis of fibronectin as a component of the ECM is increased in myofibroblasts, in comparison to fibroblasts. It was already confirmed that TGF-β is expressed in primary human ocular fibroblasts subpopulations [4], and that it plays a key role in ocular wound healing and scarring processes [53]. Following incubation of hTFs with TGF-β1, we confirmed increase in α-SMA and fibronectin expression, indicating fibroblast transformation into fibrotic active myofibroblasts. We examined the antifibrotic effect of JM, first on hTFs that are not fibrotic (without TGF-β1 stimulation), observing that with increasing concentrations of JM, up to 50 µg ml$^{-1}$, expression of fibronectin was gradually decreasing, and demonstrating the supressing effect of JM on synthesis of ECM components, in hTFs not stimulated with TGF-β1. Additionally, we did not observe any expression of α-SMA which confirms that JM did not induce fibroblast transformation into α-SMA-positive cells (which may be expected to closely resemble myofibroblasts).

We further examined how increasing doses of JM affect the expression of fibronectin and α-SMA in TGF-β1 stimulated hTFs, observing that both, fibronectin and α-SMA expression, decreased with higher values of JM. The drug-induced decrease of fibrotic cells thus leads to a decrease in their previously increased synthesis of ECM components. We propose that an additional inhibitory effect of JM on the synthesis rates of ECM components then results in a normalization of the ECM synthesis values, which are then comparable to the control samples. Therefore, JM triggers a desired antifibrotic effect in 'fibrotic' ocular fibroblasts in cell culture *in vitro*, stimulated by TGF-β1, by inhibiting fibrotic key components which are involved in scar formation processes *in vivo* [54]. Further, Western blot analyses of hTF cell lysates were performed, based on control cells and on cells stimulated with TGF-β1, and incubated with the combination of TGF-β1 and JM in increasing concentrations, yielding a strong and significant reduction of α-SMA expression. Fibronectin quantity was also affected and decreased with increasing concentrations of JM, confirming our immunofluorescence results, that JM has a notable impact specifically on the TGF-β1-induced process of fibrosis *in vitro*. Additionally, we only observed irregular differences in collagen I and collagen VI expression indicating an impact of JM on collagen synthesis which, however, could not counteract the fibrotic processes induced by TGF-β1 in total.

To conclude, JM is able to decrease synthesis of proteins relevant in fibrosis in hTFs and also in activated hTF-derived myofibroblasts *in vitro*. Therefore, it may be a promising candidate for the treatment of fibrosis following glaucoma filtration surgery or drainage device implantation *in vivo*, and thus be effective in clinical terms in preventing fibrosis after trabeculectomy surgery.

Ethics. This study was approved by the ethics committee of the University of Rostock (approval ID: A 2011 11) and followed the guidelines of the Declaration of Helsinki. Primary cultures of human Tenon fibroblasts were prepared after child strabismus surgery (Department of Ophthalmology, University of Rostock, Germany) after obtaining of their parent's or legal guardian's informed consent in writing.

Data accessibility. All data are provided as additional files. NGS data were uploaded at https://www.ebi.ac.uk/ena/data/view/PRJEB38998.

Authors' contributions. T.S., B.G.-D., and G.F. wrote the manuscript. B.G.-D. and T.S. prepared the figures describing experimental data. B.G.-D., T.S. and A.J. conducted laboratory studies. B.G.-D., T.S., S.S., A.J., I.B. and G.F. analysed the data and provided critical feedback. T.S., O.S., R.R., A.J. and G.F. conceived the study concept and procured funding for the project. All authors reviewed the manuscript and approved the final version.

Competing interests. T.S., A.J., S.S. and G.F. are listed as inventors in a pending patent application on the use of Josamycin as an antifibrotic compound, submitted in behalf of the Rostock University Medical Center. The authors declare no other competing financial interests.

Funding. This work was supported by the BMBF, VIP/VIP+ - Validierung des technologischen und gesellschaftlichen Innovationspotentials wissenschaftlicher Forschung (03V0396/03VP06230) and the EU ('Aging with elegans', Grant agreement no. 633589).

Acknowledgements. We thank Colette Leyh for excellent technical assistance.

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
