## [Reviewer comments · Royal Society Open Science]

Review History

RSOS-200441.R0 (Original submission)

Review form: Reviewer 1

Is the manuscript scientifically sound in its present form?

Yes

Are the interpretations and conclusions justified by the results?

Yes

Is the language acceptable?

Yes

Do you have any ethical concerns with this paper?

No

Have you any concerns about statistical analyses in this paper?

No

Recommendation?

Major revision is needed (please make suggestions in comments)

Comments to the Author(s)

Review RSOS-200441

Title: Suppression of the TGF- β pathway by a macrolide antibiotic decreases extracellular matrix deposition in ocular fibroblasts in vitro

In the manuscript the effects of the antibiotic Josamycin are studied on ocular fibroblasts with respect to the fibrotic response mediated by these fibroblasts. This again is linked to the ultimate use of biomedical devices (glaucoma drainage devices) in the eye which functioning suffers from the fibrotic response. The choice for this antibiotic was based on Connectivity Map data with respect to a number of genes to be involved in fibrosis.

The transforming effect of TGFbeta1 on fibroblasts towards myofibroblast is well known. The hallmarks of fibrosis also are well known. This study seems to focus solely on this particular antibiotic as it is demonstrated that it suppresses the TGF-beta pathway induced fibrotic response of the ocular fibroblasts. As a reviewer I like the described process of searching for a known drug (Josamycin goes way back in the literature) to establish a possible use as an anti-fibrotic drug. That is the true strength of this paper. The underlying documentation for the decision making has been supplied in line with the demands posed by RSOS. It is also interesting to note that the rankings shown in Tables 1 and 2 are mostly related to two cell lines (HL60 and MCF7) that may have not too much of a resemblance with the primary fibroblasts in this manuscript.

I have the following comments:

1. Title: extracellular matrix deposition BY ocular fibroblasts
2. One thing is not particularly clear: It is stated that the in silico screening involved small molecules from the CMap database that may reverse the gene expression changes caused by fibrosis. In the in vitro experiments indeed Josamycin had been added after a 48 hour stimulation with TGFbeta1. This would point at a reversal of fibrosis. In the results section, however, this reads as: "revealing a set of mechanisms that may be involved in fibrosis, and in its inhibition by JM". So is Josamycin inhibiting fibrosis or reversing it? Or are the authors referring to the same process? With respect to this I also want to point to Figure 4: "The incubation of hTFs with increasing doses of JM [5 μ g/ml, 10 μ g/ml, 25 μ g/ml, 50 μ g/ml] for 24 h resulted in a concentration-dependent decrease of fibronectin expression, demonstrating an inhibitory effect on ECM synthesis (Fig. 4)". I assume that this was without TGF stimulation. How does this result then fit in the discussion about reversal or inhibition of fibrosis?
3. Figure 1: "Overall, these upregulations reflect the buildup of extracellular matrix (ECM) in connection with the cellular actin cytoskeleton". Is this an assumption (it would be a logical one) or is this also present in this figure?
4. I feel that the representation of the TGF and Josamycin stimulations in Figures 4-7 can be reduced to two figures, as there seems to be redundancy. In Fig 7A the text accompanying the images could be inserted as has been done in figures 5 and 6.
5. Discussion: As it is now the discussion becomes too much of a repetition of the different steps taken by the authors, which are already listed in the results section.
6. I am not totally convinced that the title grasps the content of the manuscript. Extracellular matrix deposition mainly concerns measurements of fibronectin, alfaSMA does not count as an ECM. The measurement of two collagens with western blotting is not significant enough to warrant this part in the title.

Review form: Reviewer 2

Is the manuscript scientifically sound in its present form?

Yes

Are the interpretations and conclusions justified by the results?

Yes

Is the language acceptable?

Yes

Do you have any ethical concerns with this paper?

Yes

Have you any concerns about statistical analyses in this paper?

No

Recommendation?

Major revision is needed (please make suggestions in comments)

Comments to the Author(s)

Dear Author,

I have highlighted and commented my considerations in the attached PDF (Appendix A) of the manuscript. You should be able to view the comments/suggestions double clicking the highlighted text.

Decision letter (RSOS-200441.R0)

Dear Professor Fuellen,

The editors assigned to your paper ("Suppression of the TGF- β pathway by a macrolide antibiotic decreases extracellular matrix deposition in ocular fibroblasts in vitro") have now received comments from reviewers. We would like you to revise your paper in accordance with the referee and Associate Editor suggestions which can be found below (not including confidential reports to the Editor). Please note this decision does not guarantee eventual acceptance.

Please submit a copy of your revised paper before 28-Jun-2020. Please note that the revision deadline will expire at 00.00am on this date. If we do not hear from you within this time then it will be assumed that the paper has been withdrawn. In exceptional circumstances, extensions may be possible if agreed with the Editorial Office in advance. We do not allow multiple rounds of revision so we urge you to make every effort to fully address all of the comments at this stage. If deemed necessary by the Editors, your manuscript will be sent back to one or more of the original reviewers for assessment. If the original reviewers are not available, we may invite new reviewers.

When submitting your revised manuscript, you must respond to the comments made by the referees and upload a file "Response to Referees" in "Section 6 - File Upload". Please use this to

document how you have responded to the comments, and the adjustments you have made. In order to expedite the processing of the revised manuscript, please be as specific as possible in your response.

- Data accessibility

If you wish to submit your supporting data or code to Dryad (<http://datadryad.org/>), or modify your current submission to dryad, please use the following link:
<http://datadryad.org/submit?journalID=RSOS&manu=RSOS-200441>

- Competing interests

- Authors' contributions

- Acknowledgements

- Funding statement

on behalf of Dr Andrew Angel (Associate Editor) and Catrin Pritchard (Subject Editor)
openscience@royalsociety.org

Associate Editor's comments (Dr Andrew Angel):
Associate Editor: 1

Comments to the Author:

The reviewers have both recommended that major revisions are necessary. I am therefore recommending this manuscript be revised in line with the reviewers points.

Associate Editor: 2

Comments to the Author:

To the best of my knowledge the manuscript appears to be scientifically sound. My initial recommendation

Comments to Author:

Reviewers' Comments to Author:
Reviewer: 1

Comments to the Author(s)

Review RSOS-200441

Title: Suppression of the TGF- β pathway by a macrolide antibiotic decreases extracellular matrix deposition in ocular fibroblasts in vitro

In the manuscript the effects of the antibiotic Josamycin are studied on ocular fibroblasts with respect to the fibrotic response mediated by these fibroblasts. This again is linked to the ultimate use of biomedical devices (glaucoma drainage devices) in the eye which functioning suffers from the fibrotic response. The choice for this antibiotic was based on Connectivity Map data with respect to a number of genes to be involved in fibrosis.

The transforming effect of TGF β 1 on fibroblasts towards myofibroblast is well known. The hallmarks of fibrosis also are well known. This study seems to focus solely on this particular antibiotic as it is demonstrated that it suppresses the TGF-beta pathway induced fibrotic response of the ocular fibroblasts. As a reviewer I like the described process of searching for a known drug (Josamycin goes way back in the literature) to establish a possible use as an anti-fibrotic drug. That is the true strength of this paper. The underlying documentation for the decision making has been supplied in line with the demands posed by RSOS. It is also interesting to note that the rankings shown in Tables 1 and 2 are mostly related to two cell lines (HL60 and MCF7) that may have not too much of a resemblance with the primary fibroblasts in this manuscript.

I have the following comments:

1. Title: extracellular matrix deposition BY ocular fibroblasts
2. One thing is not particularly clear: It is stated that the in silico screening involved small molecules from the CMap database that may reverse the gene expression changes caused by

fibrosis. In the in vitro experiments indeed Josamycin had been added after a 48 hour stimulation with TGFbeta1. This would point at a reversal of fibrosis. In the results section, however, this reads as: “revealing a set of mechanisms that may be involved in fibrosis, and in its inhibition by JM”. So is Josamycin inhibiting fibrosis or reversing it? Or are the authors referring to the same process? With respect to this I also want to point to Figure 4: “The incubation of hTFs with increasing doses of JM [5 μ g/ml, 10 μ g/ml, 25 μ g/ml, 50 μ g/ml] for 24 h resulted in a concentration-dependent decrease of fibronectin expression, demonstrating an inhibitory effect on ECM synthesis (Fig. 4)”. I assume that this was without TGF stimulation. How does this result then fit in the discussion about reversal or inhibition of fibrosis?

3. Figure 1: “Overall, these upregulations reflect the buildup of extracellular matrix (ECM) in connection with the cellular actin cytoskeleton”. Is this an assumption (it would be a logical one) or is this also present in this figure?

4. I feel that the representation of the TGF and Josamycin stimulations in Figures 4-7 can be reduced to two figures, as there seems to be redundancy. In Fig 7A the text accompanying the images could be inserted as has been done in figures 5 and 6.

5. Discussion: As it is now the discussion becomes too much of a repetition of the different steps taken by the authors, which are already listed in the results section.

6. I am not totally convinced that the title grasps the content of the manuscript. Extracellular matrix deposition mainly concerns measurements of fibronectin, alfaSMA does not count as an ECM. The measurement of two collagens with western blotting is not significant enough to warrant this part in the title.

Reviewer: 2

Comments to the Author(s)

Dear Author,

I have highlighted and commented my considerations in the attached PDF of the manuscript. You should be able to view the comments/suggestions double clicking the highlighted text.

Author's Response to Decision Letter for (RSOS-200441.R0)

See Appendix B.

RSOS-200441.R1 (Revision)

Review form: Reviewer 1

Is the manuscript scientifically sound in its present form?

Yes

Are the interpretations and conclusions justified by the results?

Yes

Is the language acceptable?

Yes

Do you have any ethical concerns with this paper?

No

Have you any concerns about statistical analyses in this paper?

No

Recommendation?

Accept as is

Comments to the Author(s)

The authors have adequately addressed most of the reviewer comments. I do not think they significantly reduced the discussion as they stated in their rebuttal.

Decision letter (RSOS-200441.R1)

Dear Professor Fuellen,

It is a pleasure to accept your manuscript entitled "Suppression of the TGF- β pathway by a macrolide antibiotic decreases fibrotic responses by ocular fibroblasts in vitro" in its current form for publication in Royal Society Open Science. The comments of the reviewer(s) who reviewed your manuscript are included at the foot of this letter.

Kind regards,

Anita Kristiansen

Editorial Coordinator

on behalf of Dr Andrew Angel (Associate Editor) and Catrin Pritchard (Subject Editor)
openscience@royalsociety.org

Associate Editor Comments to Author (Dr Andrew Angel):

Comments to the Author:

I think that the majority of points raised by the reviewers have been adequately addressed. Therefore, I am recommending that the manuscript be accepted in its current form.

Reviewer comments to Author:

Reviewer: 1

Comments to the Author(s)

The authors have adequately addressed most of the reviewer comments. I do not think they significantly reduced the discussion as they stated in their rebuttal.

Appendix A**ROYAL SOCIETY
OPEN SCIENCE****Suppression of the TGF- β pathway by a macrolide antibiotic
decreases extracellular matrix deposition in ocular
fibroblasts in vitro**

Journal:	Royal Society Open Science
Manuscript ID	RSOS-200441
Article Type:	Research
Date Submitted by the Author:	19-Mar-2020
Complete List of Authors:	Stahnke, Thomas; Rostock University Medical Center Gajda-Derylo, Beata; Medical University of Lublin Jünemann, Anselm; Rostock University Medical Center Stachs, Oliver; Rostock University Medical Center Sterenczak, Katharina A.; Rostock University Medical Center Rejdak, Robert; Medical University of Lublin Beck, Julia; Chronix Biomedical GmbH Schütz, Ekkehard; Chronix Biomedical GmbH Möller, Steffen; Rostock University Medical Center Barrantes, Israel; Rostock University Medical Center Warsow, Gregor; Rostock University Medical Center Struckmann, Stepahn; Rostock University Medical Center Fuellen, Georg; Rostock University Medical Center, Institute for Biostatistics and Informatics in Medicine and Ageing Research
Subject:	bioinformatics < BIOLOGY, cellular biology < BIOLOGY
Keywords:	Fibrosis, Drug Repositioning, Glaucoma Surgery
Subject Category:	Biochemistry, Cellular and Molecular Biology

Author-supplied statements

Relevant information will appear here if provided.

Ethics

Does your article include research that required ethical approval or permits?:

This article does not present research with ethical considerations

Statement (if applicable):

CUST_IF_YES_ETHICS :No data available.

Data

It is a condition of publication that data, code and materials supporting your paper are made publicly available. Does your paper present new data?:

Yes

Statement (if applicable):

All data are provided as additional files, see the last page of the manuscript, which describes the ESM, and provides links to the other data that are, in part, very large in size so that they could not be submitted via the submission system.

Conflict of interest

I/We declare a competing interest

Statement (if applicable):

T.S., A.J., S.S. and G.F. are listed as inventors in a pending patent application on the use of Josamycin as an anti-fibrotic compound, submitted in behalf of the Rostock University Medical Center. The authors declare no other competing financial interests.

Authors' contributions

This paper has multiple authors and our individual contributions were as below

Statement (if applicable):

T.S., B.G.-D., and G.F. wrote the manuscript. B.G.-D. and T.S. prepared the figures describing experimental data. B.G.-D., T.S. and A.J. conducted laboratory studies. B.G.-D., T.S., S.S., A.J., I.B. and G.F. analyzed the data and provided critical feedback. T.S., O.S., R.R., A.J. and G.F. conceived the study concept and procured funding for the project. All authors reviewed the manuscript and approved the final version.

Suppression of the TGF- β pathway by a macrolide antibiotic decreases extracellular matrix deposition in ocular fibroblasts *in vitro*

Thomas Stahnke¹, Beata Gajda-Deryło², Anselm G. Jünemann¹, Oliver Stachs¹, Katharina A. Sterenczak¹, Robert Rejdak², Julia Beck³, Ekkehard Schütz³, Steffen Möller⁴, Israel Barrantes⁴, Gregor Warsow^{4,5}, Stephan Struckmann^{4,6,*}, Georg Fuellen^{4,*}

¹ Department of Ophthalmology, Rostock University Medical Center, Rostock, Germany

² Department of General Ophthalmology, Medical University in Lublin, Poland

³ Chronix Biomedical GmbH, Göttingen, Germany

⁴ Institute for Biostatistics and Informatics in Medicine and Ageing Research, Rostock University Medical Center, Rostock, Germany

⁵ current address: Omics IT and Data Management Core Facility, German Cancer Research Center (DKFZ), Heidelberg, Germany.

⁶ SHIP-KEF, Institute for Community Medicine, Greifswald University Medical Center, Greifswald, Germany

* Correspondence and requests for materials should be addressed to G.F. (email: fuellen@uni-rostock.de) or S.S. (email: stephan.struckmann@uni-rostock.de)

Abstract

[revised manuscript text omitted]

Fig. 3: Relative viability (**A**) and relative proliferation (**B**) of hTFs in response to JM *in vitro*. hTFs were treated with JM in concentrations as indicated. CellQuanti-Blue-assay was carried out to quantify relative cell viability. Cell Proliferation ELISA, BrdU (chemiluminescence)-assay was carried out to quantify relative proliferation. Data are presented as mean \pm SD. The results represent the means of three independent experiments. Levels of significance: * $p \leq 0.05$; ** $p \leq 0.01$; *** $p \leq 0.001$.

The incubation of hTFs with increasing doses of JM [5μg/ml, 10μg/ml, 25μg/ml, 50μg/ml] for 24 h resulted in a concentration-dependent decrease of fibronectin expression, demonstrating an inhibitory effect on ECM synthesis (Fig. 4).

Fig. 4: **A:** Immunofluorescence of fibronectin (red) in hTFs treated with increasing JM concentrations: JM5 [5μg/ml, 6*10⁻⁹ mol/ml], JM10 [10μg/ml, 12*10⁻⁹ mol/ml], JM25 [25μg/ml, 30*10⁻⁹ mol/ml], JM50 [50μg/ml,

60*10⁻⁹ mol/ml]. Bar represents 10 μm. **B:** Quantification of fibronectin expression after JM inhibition in hTFs. Each column represents the mean ± SD from three independent experiments. The asterisk (*) indicates significance obtained by comparison of JM5, JM10, JM25 and JM50 to untreated cultures. Levels of significance: *p≤0.05; **p≤0.01; ***p≤0.001.

[revised manuscript text omitted]

JM concentration of 150 μ g/ml. This finding demonstrates that JM triggers a desired
antifibrotic effect in “fibrotic” ocular fibroblasts in cell culture *in vitro*, stimulated by TGF-
β 1, by inhibiting fibrotic key components which are involved in scar formation processes
*in vivo* [53].

To confirm our results, Western blot analyses of hTF cell lysates were performed,
based on control cells and on cells stimulated with TGF- β 1, and incubated with the
combination of TGF- β 1 and JM in increasing concentrations. We observed a strong and
significant reduction of α -SMA expression. Fibronectin quantity was also affected and
decreased with increasing concentrations of JM. These findings confirm our
immunofluorescence results, showing that JM has a notable impact specifically on the
TGF- β 1-induced process of fibrosis *in vitro*. Additionally, we only observed irregular
differences in collagen I and collagen VI expression indicating an impact of JM on
collagen synthesis which, however, could not counteract the fibrotic processes induced
by TGF- β 1 in total.

To conclude, JM is able to decrease synthesis of proteins relevant in fibrosis in
hTFs and also in activated hTF-derived myofibroblasts *in vitro*. Therefore, it may be a
promising candidate for the treatment of fibrosis following glaucoma filtration surgery or
drainage device implantation *in vivo*, and thus be effective in clinical terms in preventing
fibrosis after trabeculectomy surgery.

Bibliography

- 1. Quigley, H.A., *Glaucoma*. Lancet, 2011. **377**(9774): p. 1367-77.
- 2. Schwartz, M., *Neurodegeneration and neuroprotection in glaucoma: development of a*
*therapeutic neuroprotective vaccine: the Friedenwald lecture*. Invest Ophthalmol Vis
Sci, 2003. **44**(4): p. 1407-11.
- 3. Khaw, P.T., et al., *Enhanced trabeculectomy: the Moorfields Safer Surgery System*. Dev
Ophthalmol, 2012. **50**: p. 1-28.
- 4. Stahnke, T., et al., *Different fibroblast subpopulations of the eye: a therapeutic target to*
*prevent postoperative fibrosis in glaucoma therapy*. Exp Eye Res, 2012. **100**: p. 88-97.
- 5. Hengerer, F.H., et al., *Second-Generation Trabecular Micro-Bypass Stents as*
*Standalone Treatment for Glaucoma: A 36-Month Prospective Study*. Adv Ther, 2019.
**36**(7): p. 1606-1617.
- 6. Ehrnrooth, P., et al., *Long-term outcome of trabeculectomy in terms of intraocular*
*pressure*. Acta Ophthalmol Scand, 2002. **80**(3): p. 267-71.
- 7. Hong, C.H., et al., *Glaucoma drainage devices: a systematic literature review and*
*current controversies*. Surv Ophthalmol, 2005. **50**(1): p. 48-60.
- 8. Gajda-Derylo, B., et al., *Comparison of cytokine/chemokine levels in aqueous humor of*
*primary open-angle glaucoma patients with positive or negative outcome following*
*trabeculectomy*. Biosci Rep, 2019. **39**(5).
- 9. Schlunck, G., et al., *Conjunctival fibrosis following filtering glaucoma surgery*. Exp
Eye Res, 2016. **142**: p. 76-82.

10. Stahnke, T., et al., *Suppression of TGF-beta pathway by pirfenidone decreases extracellular matrix deposition in ocular fibroblasts in vitro*. PLoS One, 2017. **12**(2): p. e0172592.
11. Wynn, T.A., *Cellular and molecular mechanisms of fibrosis*. J Pathol, 2008. **214**(2): p. 199-210.
12. Green, E., et al., *5-Fluorouracil for glaucoma surgery*. Cochrane Database Syst Rev, 2014(2): p. CD001132.
13. Cabourne, E., et al., *Mitomycin C versus 5-Fluorouracil for wound healing in glaucoma surgery*. Cochrane Database Syst Rev, 2015(11): p. CD006259.
14. Yu-Wai-Man, C. and P.T. Khaw, *Developing novel anti-fibrotic therapeutics to modulate post-surgical wound healing in glaucoma: big potential for small molecules*. Expert Rev Ophthalmol, 2015. **10**(1): p. 65-76.
15. Lamb, J., et al., *The Connectivity Map: using gene-expression signatures to connect small molecules, genes, and disease*. Science, 2006. **313**(5795): p. 1929-35.
16. Stahnke, T., et al., [*Systems Biology in Ophthalmology - Innovative Drug Identification for the Specific Prevention of Postoperative Fibrosis*]. Klin Monbl Augenheilkd, 2019. **236**(12): p. 1428-1434.
17. Gassmann, M., et al., *Quantifying Western blots: pitfalls of densitometry*. Electrophoresis, 2009. **30**(11): p. 1845-55.
18. Mortazavi, A., et al., *Mapping and quantifying mammalian transcriptomes by RNA-Seq*. Nat Methods, 2008. **5**(7): p. 621-8.
19. Ernst, M., et al., *FocusHeuristics - expression-data-driven network optimization and disease gene prediction*. Sci Rep, 2017. **7**: p. 42638.
20. Franceschini, A., et al., *STRING v9.1: protein-protein interaction networks, with increased coverage and integration*. Nucleic Acids Res, 2013. **41**(Database issue): p. D808-15.
21. Liu, B., *Web Data Mining: Exploring Hyperlinks, Contents, and Usage Data*. 2011: Springer-Verlag, Berlin / Heidelberg.
22. Warsow, G., et al., *ExprEssence--revealing the essence of differential experimental data in the context of an interaction/regulation network*. BMC Syst Biol, 2010. **4**: p. 164.
23. Subramanian, A., et al., *A Next Generation Connectivity Map: L1000 Platform and the First 1,000,000 Profiles*. Cell, 2017. **171**(6): p. 1437-1452 e17.
24. Barth, P.J. and C.C. Westhoff, *CD34+ fibrocytes: morphology, histogenesis and function*. Curr Stem Cell Res Ther, 2007. **2**(3): p. 221-7.
25. Jimenez-Heffernan, J.A., et al., *Immunohistochemical characterization of fibroblast subpopulations in normal peritoneal tissue and in peritoneal dialysis-induced fibrosis*. Virchows Arch, 2004. **444**(3): p. 247-56.
26. Yu-Wai-Man, C., et al., *Genome-wide RNA-Sequencing analysis identifies a distinct fibrosis gene signature in the conjunctiva after glaucoma surgery*. Sci Rep, 2017. **7**(1): p. 5644.
27. Seong, G.J., et al., *TGF-beta-induced interleukin-6 participates in transdifferentiation of human Tenon's fibroblasts to myofibroblasts*. Mol Vis, 2009. **15**: p. 2123-8.
28. Andreev, K., et al., *Expression of bone morphogenetic proteins (BMPs), their receptors, and activins in normal and scarred conjunctiva: role of BMP-6 and activin-A in conjunctival scarring?* Exp Eye Res, 2006. **83**(5): p. 1162-70.
29. Schafer, S., et al., *IL-11 is a crucial determinant of cardiovascular fibrosis*. Nature, 2017. **552**(7683): p. 110-115.
30. Cook, S.A., et al., *IL-11 is a therapeutic target in idiopathic pulmonary fibrosis*. 597 (2018). . bioRxiv, 2018: p. doi: 10.1101/336537.
31. Huang, W., et al., *Coronin 2A mediates actin-dependent de-repression of inflammatory response genes*. Nature, 2011. **470**(7334): p. 414-8.
32. Agarwal, P., et al., *Enhanced deposition of cartilage oligomeric matrix protein is a common feature in fibrotic skin pathologies*. Matrix Biol, 2013. **32**(6): p. 325-31.

- 33. Vuga, L.J., et al., *Cartilage oligomeric matrix protein in idiopathic pulmonary fibrosis*. PLoS One, 2013. **8**(12): p. e83120.
- 34. Andreasson, K., et al., *Cartilage oligomeric matrix protein: a new promising biomarker* of liver fibrosis in chronic hepatitis C. *Infect Dis (Lond)*, 2015. **47**(12): p. 915-8.
- 35. Desmouliere, A., et al., *Transforming growth factor-beta 1 induces alpha-smooth* muscle actin expression in granulation tissue myofibroblasts and in quiescent and
- *growing cultured fibroblasts*. *J Cell Biol*, 1993. **122**(1): p. 103-11.
- 36. *Fluorouracil Filtering Surgery Study one-year follow-up. The Fluorouracil Filtering*
- *Surgery Study Group*. *Am J Ophthalmol*, 1989. **108**(6): p. 625-35.
- 37. Cheung, J.C., et al., *Intermediate-term outcome of variable dose mitomycin C filtering*
- *surgery*. *Ophthalmology*, 1997. **104**(1): p. 143-9.
- 38. Stamper, R.L., M.G. McMenemy, and M.F. Lieberman, *Hypotonous maculopathy after*
- *trabeculectomy with subconjunctival 5-fluorouracil*. *Am J Ophthalmol*, 1992. **114**(5): p.
- 544-53.
- 39. Watson, P.G., et al., *The complications of trabeculectomy (a 20-year follow-up)*. *Eye*
- *(Lond)*, 1990. **4 (Pt 3)**: p. 425-38.
- 40. Chen, H.J., et al., *Efficacy and Safety of Bevacizumab Combined with Mitomycin C or*
- *5-Fluorouracil in Primary Trabeculectomy: A Meta-Analysis of Randomized Clinical*
- *Trials*. *Ophthalmic Res*, 2018. **59**(3): p. 155-163.
- 41. Nilforushan, N., et al., *Subconjunctival bevacizumab versus mitomycin C adjunctive to*
- *trabeculectomy*. *Am J Ophthalmol*, 2012. **153**(2): p. 352-357 e1.
- 42. Auzou, M., et al., *In vitro activity of josamycin against Streptococcus pyogenes isolated*
- *from patients with upper respiratory tract infections in France*. *Med Mal Infect*, 2015.
- **45**(7): p. 293-6.
- 43. Lozano, R. and A. Balaguer, *Josamycin in the treatment of bronchopulmonary*
- *infections*. *Clin Ther*, 1991. **13**(2): p. 281-8.
- 44. Kanai, K., et al., *Suppression of matrix metalloproteinase production from nasal*
- *fibroblasts by macrolide antibiotics in vitro*. *Eur Respir J*, 2004. **23**(5): p. 671-8.
- 45. Nonaka, M., et al., *A macrolide antibiotic, roxithromycin, inhibits the growth of nasal*
- *polyp fibroblasts*. *Am J Rhinol*, 1999. **13**(4): p. 267-72.
- 46. Park, H.H., et al., *The effect of macrolides on myofibroblast differentiation and collagen*
- *production in nasal polyp-derived fibroblasts*. *Am J Rhinol Allergy*, 2010. **24**(5): p.
- 348-53.
- 47. Yatsunami, J., et al., *Antiangiogenic and antitumor effects of 14-membered ring*
- *macrolides on mouse B16 melanoma cells*. *Clin Exp Metastasis*, 1999. **17**(4): p. 361-7.
- 48. Morikawa, K., et al., *Immunomodulatory effects of three macrolides, midecamycin*
- *acetate, josamycin, and clarithromycin, on human T-lymphocyte function in vitro*.
- *Antimicrob Agents Chemother*, 1994. **38**(11): p. 2643-7.
- 49. Alileche, A., et al., *IL-2 production by myofibroblasts from post-radiation fibrosis in*
- *breast cancer patients*. *Int Immunol*, 1994. **6**(10): p. 1585-91.
- 50. Hubeau, C., et al., *Dysregulation of IL-2 and IL-8 production in circulating T*
- *lymphocytes from young cystic fibrosis patients*. *Clin Exp Immunol*, 2004. **135**(3): p.
- 528-34.
- 51. Torr, E.E., et al., *Myofibroblasts exhibit enhanced fibronectin assembly that is intrinsic*
- *to their contractile phenotype*. *J Biol Chem*, 2015. **290**(11): p. 6951-61.
- 52. Van de Velde, S., et al., *Modulation of wound healing in glaucoma surgery*. *Prog Brain*
- *Res*, 2015. **221**: p. 319-40.
- 53. Xue, M. and C.J. Jackson, *Extracellular Matrix Reorganization During Wound Healing*
- *and Its Impact on Abnormal Scarring*. *Adv Wound Care (New Rochelle)*, 2015. **4**(3): p.
- 119-136.

Acknowledgments

We thank Colette Leyh for excellent technical assistance.

Funding

This work was supported by the BMBF, VIP - Validierung des Innovationspotentials wissenschaftlicher Forschung (03V0396) and the EU ("Aging with elegans", Grant agreement No 633589).

Author Contributions

T.S., B.G.-D., and G.F. wrote the manuscript. B.G.-D. and T.S. prepared the figures describing experimental data. B.G.-D., T.S. and A.J. conducted laboratory studies. B.G.-D., T.S., S.S., A.J., I.B. and G.F. analyzed the data and provided critical feedback. T.S., O.S., R.R., A.J. and G.F. conceived the study concept and procured funding for the project. All authors reviewed the manuscript and approved the final version.

Additional Information

Competing financial interests: T.S., A.J., S.S. and G.F. are listed as inventors in a pending patent application on the use of Josamycin as an anti-fibrotic compound, submitted in behalf of the Rostock University Medical Center. The authors declare no other competing financial interests.

Electronic supplementary material

Electronic supplementary material 1 (Fibrosis Pearson correlations.xlsx). Pearson correlation-based ranking for the top 500 drugs, considering similarity of gene expression changes with the query describing fibrosis, and starting point of further analyses.

Electronic supplementary material 2 (Fibrosis.cys). The networks filtered by the FocusHeuristics, for fibrosis, including the gene expression data mapped onto these.

Electronic supplementary material 3 (JM.cys). The networks filtered by the FocusHeuristics, for JM, including the gene expression data mapped onto these.

The CMap-based R data frame and the code used to generate it can be found at <https://bitbucket.org/ibima/cmapreanalysed2016>.

The igraph representation of the original STRING network is provided at <https://bitbucket.org/ibima/fibrosisdrug2020> as STRING10HUMANDNW_BIG.RDATA (33975488 bytes). At the same location, the code used to calculate the Jaccard index is provided as jacc.correlation.R.

Appendix B

Dear Dr Dunn, dear Dr Angel, dear Dr Pritchard,

Thanks a lot for handling the manuscript; below please find our response to the comments.

>> Associate Editor's comments (Dr Andrew Angel):

>> Associate Editor: 1

>>

>> Comments to the Author:

>>

>> The reviewers have both recommended that major revisions are necessary. I

>> am therefore recommending this manuscript be revised in line with the

>> reviewers points.

Thanks for giving us the opportunity to revise the manuscript!

>> Associate Editor: 2

>>

>> Comments to the Author:

>>

>> To the best of my knowledge the manuscript appears to be scientifically

>> sound. My initial recommendation

>>

>> Comments to Author:

>>

>> Reviewers' Comments to Author:

>> Reviewer: 1

>>

>> Comments to the Author(s)

>>

>> Review RSOS-200441

>> Title: Suppression of the TGF- β pathway by a macrolide antibiotic decreases

>> extracellular matrix deposition in ocular fibroblasts in vitro

>> In the manuscript the effects of the antibiotic Josamycin are studied on

>> ocular fibroblasts with respect to the fibrotic response mediated by these

>> fibroblasts. This again is linked to the ultimate use of biomedical devices

>> (glaucoma drainage devices) in the eye which functioning suffers from the

>> fibrotic response. The choice for this antibiotic was based on Connectivity

>> Map data with respect to a number of genes to be involved in fibrosis.

>> The transforming effect of TGFbeta1 on fibroblasts towards myofibroblast is

>> well known. The hallmarks of fibrosis also are well known. This study

>> seems to focus solely on this particular antibiotic as it is demonstrated

>> that it suppresses the TGF-beta pathway induced fibrotic response of the

>> ocular fibroblasts. As a reviewer I like the described process of searching

>> for a known drug (Josamycin goes way back in the literature) to establish

>> a possible use as an anti-fibrotic drug. That is the true strength of this

>> paper. The underlying documentation for the decision making has been

>> supplied in line with the demands posed by RSOS. It is also interesting to

>> note that the rankings shown in Tables 1 and 2 are mostly related to two

>> cell lines (HL60 and MCF7) that may have not too much of a resemblance with

>> the primary fibroblasts in this manuscript.

>> I have the following comments:

>> 1. Title: extracellular matrix deposition BY ocular fibroblasts

We agree, and we modified the title. It now reads "Suppression of the TGF- β pathway by a macrolide antibiotic decreases fibrotic responses by ocular fibroblasts in vitro".

>> 2. One thing is not particularly clear: It is stated that the in silico

>> screening involved small molecules from the CMap database that may reverse

>> the gene expression changes caused by fibrosis. In the in vitro experiments
>> indeed Josamycin had been added after a 48 hour stimulation with
>> TGFbeta1. This would point at a reversal of fibrosis. In the results
>> section, however, this reads as: “revealing a set of mechanisms that may be
>> involved in fibrosis, and in its inhibition by JM”. So is Josamycin
>> inhibiting fibrosis or reversing it ? Or are the authors referring to the
>> same process? With respect to this I also want to point to Figure 4: “The
>> incubation of hTFs with increasing doses of JM [5µg/ml, 10µg/ml, 25µg/ml,
>> 50µg/ml] for 24 h resulted in a concentration-dependent decrease of
>> fibronectin expression, demonstrating an inhibitory effect on ECM synthesis
>> (Fig. 4)”. I assume that this was without TGF stimulation. How does this result
>> then fit in the discussion about reversal or inhibition of fibrosis ?

Our results show that JM generally exerts an inhibitory influence on the expression rate of ECM components. Since JM also lowers the expression of α -SMA, a reversal of the fibrotic processes is likely. Thus, the drug-induced decrease in fibrotic cells compensates for their increased synthesis of ECM components. The additional inhibitory influence on the ECM synthesis then leads to expression rates that are similar to those of the control samples. For clarification we have included an explanatory text passage in the Discussion:

“The drug-induced decrease of fibrotic cells thus leads to a decrease in their previously increased synthesis of ECM components. We propose that an additional inhibitory effect of JM on the synthesis rates of ECM components then results in a normalization of the ECM synthesis values, which are then comparable to the control samples.”

>> 3. Figure 1: “Overall, these upregulations reflect the buildup of
>> extracellular matrix (ECM) in connection with the cellular actin
>> cytoskeleton”. Is
>> this an assumption (it would be a logical one) or is this also present in
>> this figure ?

We agree and changed the sentence accordingly. The direct relationship between the actin cytoskeleton and an increased synthesis of ECM components is not depicted in the figure. However, it is known from the literature that the expression of α -SMA indicates an activation of fibroblasts and their transformation to myofibroblasts, which can also be characterized by an increased synthesis rate of ECM components:

“Overall, an increased expression rate of the actin cytoskeleton, and in particular α -SMA, indicates a transformation of fibroblasts to myofibroblasts, which are also characterized by an increased synthesis of ECM components [10, 11].”

>> 4. I feel that the representation of the TGF and Josamycin stimulations in
>> Figures 4-7 can be reduced to two figures, as there seems to be
>> redundancy. In Fig 7A the text accompanying the images could be inserted as
>> has been done in figures 5 and 6.

We share the reviewer's opinion that the figures are partially redundant. We have therefore deleted figure 4 from the manuscript, as its message is also represented in figure 5 (now the new 4). We have retained Figure 6 (now 5), as it is only here that the sole effects of JM in higher doses are represented. Furthermore, the associated quantification is also part of Figure 7B and C (now 6B and C). For clarification, we have adapted the caption to the new Figure 6. The references to the figures have been adapted to the new figure numbers throughout the manuscript.

>> 5. Discussion: As it is now the discussion becomes too much of a repetition
>> of the different steps taken by the authors, which are already listed in
>> the results section.

We agree and we shortened the Discussion text in a number of ways, see the manuscript.

>> 6. I am not totally convinced that the title grasps the content of the

>> manuscript. Extracellular matrix deposition mainly concerns measurements of
>> fibronectin, alfaSMA does not count as an ECM. The measurement of two
>> collagens with western blotting is not significant enough to warrant this
>> part in the title.

See above, we suggest the new title "Suppression of the TGF- β pathway by a macrolide antibiotic decreases fibrotic responses by ocular fibroblasts in vitro"

>> Reviewer: 2

>>

>> Comments to the Author(s)

>>

>> Dear Author,

>>

>> I have highlighted and commented my considerations in the attached PDF of
>> the manuscript. You should be able to view the comments/suggestions double
>> clicking the highlighted text.

>> 3/34 I think this sentence can be improved. I almost did not understand it at first.

Thanks; we now write "The overshooting of matrix deposits by myofibroblasts, cellular infiltrates by e.g. mast cells, and a reduced remodelling of the tissue are called fibrosis or scarring."

>> several times: use italics: pages 3,4,9,

Changed as suggested.

>> 4/37 Perhaps "maintained" would be better.

Yes, changed as suggested.

>> 4/52 I did not see this abbreviation previously in the text. Is this fetal calf serum?

Yes, we now state so.

>> 4/57 "0% FCS "

Yes, 0% FCS, we now state so.

>> 7/15 - Have you registered the NGS data into publicly available repositories such as Gene Expression

>> Omnibus or ArrayExpress?

>> - What platform was used to perform the sequencing?

>> - Describe the pre-processing and differential expression analysis of NGS data.

>> - I recommend splitting the NGS processing and analyses description and the CMap/FocusHeuristics methods description.

We uploaded the NGS data and added an extra section regarding the NGS processing and analyses as follows.

NGS processing and analyses.

Poly-A RNA was enriched from 1 μ g of total RNA and cDNA sequencing libraries were prepared using the NEBNext Ultra RNA Library Prep Kit for Illumina. Libraries were sequenced on a NextSeq500 platform (Illumina) to obtain ~50 million 75-base-single-end reads per library. The raw data are available at <https://www.ebi.ac.uk/ena/data/view/PRJEB38998>. Low quality reads were identified by FastQC-0.11.2 and were filtered using Trimmomatic-0.32. The remaining reads were mapped to the Homo sapiens Ensembl GRCh38 assembly using TopHat2 and Bowtie2 with default parameters. Finally, HTSeq-count-0.6.1 (part of the SAMtools-1.3246) was used to generate the count files based on the mapped reads.

>> several times: Change to "Table 1/2": 9(2x),10(2x),12

Fixed.

>> 9/15 Link is broken

Fixed.

>> several times: Changing this asterisk to " 6×10^{-9} " would be better.: pages 14,15,16,

Changed.

>> 18/46 The resolution of your figures are very low. I wonder if you can improve that, specially since most of your main experimental results rely on visualization (immunoflorescence and Western blot). However your plots are also hard to read with the curent figure resolution, specially the markings inside the graphs.

We prepared new figures with higher resolution.